# Ellagic Acid and Its Anti-Aging Effects on Central Nervous System

**DOI:** 10.3390/ijms231810937

**Published:** 2022-09-19

**Authors:** Heyu Zhu, Yuanmei Yan, Yi Jiang, Xianfang Meng

**Affiliations:** Department of Neurobiology, School of Basic Medical Sciences, Tongji Medical College, Huazhong University of Science and Technology, Wuhan 430030, China

**Keywords:** ellagic acid, anti-aging, anti-inflammation, antioxidant, neuroprotection

## Abstract

Aging is an unavoidable biological process that leads to the decline of human function and the reduction in people’s quality of life. Demand for anti-aging medicines has become very urgent. Many studies have shown that ellagic acid (EA), a phenolic compound widely distributed in dicotyledonous plants, has powerful anti-inflammation and antioxidant properties. Moreover, it has been demonstrated that EA can enhance neuronal viability, reduce neuronal defects, and alleviate damage in neurodegenerative diseases such as Alzheimer’s disease, Parkinson’s disease, and cerebral ischemia. This paper reviews the biochemical functions and neuroprotective effects of EA, showing the clinical value of its application.

## 1. Introduction

Aging is an unavoidable process characterized by the gradual decrease in various biological activities in the organs and the reduction in repair capacity for external and internal injuries [1,2]. Researchers are devoted to investigating the molecular mechanisms of aging in order to extend the human lifespan. Some dietary compounds in fruits and vegetables have been extracted and evaluated, and it has been proven that some phenolic substances such as ellagic acid (EA) have anti-aging effects by scavenging free radicals and exhibiting antioxidative and anti-inflammatory properties [3].

EA (Figure 1) is a chromene-dione derivative (C_14_H_6_O_8_) [4]. There are four hydroxyl groups, two lactones, and two hydrocarbon rings which give EA the ability to accept electrons from different substrates and participate in antioxidant reactions [5]. It exists in the natural environment in free form, and, more often than not, in condensed form, for example, as ellagitannins (ETs), glycosides, and so on. EA is widely present in some fruits, nuts, and seeds, for instance, pomegranates, black raspberries, raspberries, strawberries, walnuts, and almonds [6]. Several studies have revealed that EA possesses potent biochemical and biological activities, including antioxidative, anti-inflammatory, and neuroprotective effects.

EA is considered to possess higher antioxidant activity than vitamin E succinate and melatonin. It plays important roles in scavenging free radicals, enhancing the activities of antioxidant enzymes, inhibiting lipid peroxidation, and reducing ROS production [5]. Moreover, EA can also inhibit the pro-oxidant effect of metal ions through chelation reactions [7,8]. Inflammation and oxidative stress are closely related, while EA has powerful antioxidant and anti-inflammatory capabilities. There are three main signaling pathways that induce inflammatory responses, Nf-κB, MAPKs, and JAK/STAT pathways. EA can inhibit all three pathways to prevent the expression of downstream genes and thus inhibit the release of inflammatory factors such as TNF-α, IL-1β, IL-6, iNOX, and so on [9].

Due to its powerful antioxidative and anti-inflammatory properties, EA has been shown to have protective effects on several kinds of diseases of the nervous system. In some neurodegenerative diseases, for instance, Alzheimer’s disease (AD) and Parkinson’s disease (PD), and cerebral ischemia, excessive oxidative stress and inflammatory response are involved in neuronal death, brain tissue damage, and the development of the diseases. Therefore, it is of great significance to inhibit oxidative stress, scavenge free radicals, and reduce or even eliminate inflammatory responses for the treatment and prevention of nervous system diseases. Numerous pieces of research have shown that EA can significantly improve neurological pathologies through its antioxidative and anti-inflammatory properties [5,10].

In this review, we aim to comprehensively summarize the antioxidative and anti-inflammatory mechanisms of EA and its protective effects on the central nervous system in order to provide a potential therapy for the treatment of neurological disorders.

## 2. Antioxidants

Oxidative stress can be defined as a disruption of the balance between oxidants and antioxidants in the organism. Excessive free radicals causing negative effects on the body are one of the most popular theories of aging [11]. Oxidative stress has been found to be involved in chronic cardiovascular disease [12], hepatic functional changes [13], and neurological disorders [14], and is thought to be a major cause of various cancers in animals and humans [15].

Oxidants can be classified as reactive oxygen species (ROS), reactive nitrogen species (RNS) [16], and oxidized lipids [17]. ROS include superoxide anions (·O_2_−), hydroxyl radicals (·HO−), hydrogen peroxide (H_2_O_2_), etc.; RNSs include nitric oxide (·NO), nitrogen dioxide (·NO_2_), peroxynitrite (·ONOO−), etc.; and oxidized lipids include arachidonic acid, linoleic acid, etc. The antioxidant mechanisms in the body consist of the enzymatic antioxidant system, such as superoxide dismutase (SOD), catalase (CAT), glutathione peroxidase (GPx), and lactate dehydrogenase (LDH), and non-enzymatic antioxidant systems, such as vitamin C, vitamin E, melatonin, glutathione, and superoxide anion (SOA) [18].

EA plays a distinct role in antioxidation by scavenging free radicals, chelating metal ions such as iron and copper, and regulating several signaling pathways [19,20] (Figure 2). As a highly efficient antioxidant, EA can scavenge two free radicals (a ROO· and an O_2_·) per cycle in aqueous solutions of physiological pH until the intermediate products are consumed in other reactions [21]. EA has been found to scavenge hydroxyl, methoxy, and nitrogen dioxide efficiently (OH·> OCH_3_·> NO_2_·) [22]. Researchers have demonstrated that EA can also scavenge O_2_^−^ and OH· to reduce the peroxidation of lipids [23]. Additionally, EA may regulate peroxynitrite reactions to reduce neurotoxicity and cytotoxicity resulting from the excess production of ·ONOO− [24]. When deprotonated, EA can chelate copper in an aqueous solution to reduce the production of free radicals, especially OH·, in a concentration-dependent manner [21].

EA can regulate oxidative stress in several ways to protect the body. Excessive ROS can damage the human body by increasing membrane lipid, damaging DNA, and triggering apoptosis [19]. Hseu et al. showed that EA can protect human keratinocyte cells from ultraviolet A-induced oxidative stress and apoptosis by upregulating the expression of HO-1, Nrf-2, and SOD [25]. In paraquat-induced oxidative stress in human alveolar A94 cells, EA treatment reduced the release of LDH and lowered the level of NF-κB and HO-1 [26]. Moreover, EA can reduce the production of ROS via regulating antioxidants and antioxidant enzymes as well. EA supplementation has also been shown to increase the activity of antioxidant enzymes in d-galactose-induced rats, such as SOD, CAT, and GPX [27].

Although EA can reduce oxidative stress efficiently through various pathways, the available studies focusing on the effects of EA in humans are very limited and most of them are based on specific human cell lines [18]; the clinical evidence on the pharmacological properties of EA is scarce [20]. Another challenge is the low bioavailability of EA due to poor intestinal absorption [19]. Thus, it is necessary to develop more preclinical studies and look for a better method of administration.

## 3. Anti-Inflammation

Inflammation is the defensive response of living tissue and the vascular system to injury factors [28]. However, in some specific situations, inflammation can damage the host and induce the dysfunction of organs [29]. Many studies have proved that inflammation is an underlying factor in plenty of diseases, especially chronic diseases such as cardiovascular and neurodegenerative diseases, obesity, cancer, and aging [30]. External stimulus and cell damage could activate inflammatory cells and induce the activation of inflammation signal pathways including NF-κB, MAPK, and JAK-STAT pathways [9]. In addition, leukocytes will migrate to the damaged parts and induce an inflammatory response. So, it is a feasible strategy to prevent and treat chronic diseases through the inhibition of inflammation.

The nf-κB pathway plays a vital role in inflammation, immune response, and apoptosis. The activation of Nf-κB is caused by extracellular stimuli, for example, cytokines, radiation, heavy metals, and viruses [9]. Normally, IκB presents in the cytoplasm and binds to Nf-κB through the specific ankyrin repeat-containing domain at IκB’s C terminal to inhibit the transfer of Nf-κB. When the pathway is activated, IκB will be phosphorylated by IκB Kinase (IKK) so Nf-κB will be free to transfer to the nucleus to regulate genes involved in inflammation [31]. It has been found that EA can bond with one of the precursor proteins (p50) of Nf-κB, p105. This bonding obstructs the binding of Nf-κB to a specific DNA domain so the genes regulated by Nf-κB would not be expressed [32]. Another function of EA on Nf-κB is inhibiting the phosphorylation of IkB and the subsequent translocation of NF-kB into the nucleus [31]. Numerous pieces of evidence have shown that EA can inhibit the Nf-κB pathway and downregulate the expression of iNOS, TNF-α, COX-2, and IL-6 [33,34,35] (Figure 3).

The mammalian MAPK family has three constituents: ERKs, JNKs, and p38 MAP Kinases [36]. The activation of ERKs and JNKs could lead to the phosphorylation and activation of p38 which then cause an inflammatory response [37]. ERKs are usually related to the mitogens and differentiation signals while JNKs and p38 MAPKs are activated by inflammatory signals and stress stimuli [38]. It has been proved that EA can inhibit all the MAPKs’ activation in a dose-dependent manner [39]. Similarly, another group revealed that EA may be the inhibitor of these three MAPKs to downregulate some proinflammatory factors [40]. However, the potential mechanism of this inhibition process is unclear.

The JAK/STAT pathway is another significant inflammatory signal pathway and is highly conserved. The activated JAK/STAT pathway can promote the production of inflammatory cytokines which play a vital role in the inflammatory process [41]. In the cytoplasm, STAT is phosphorylated by JAK and then dimerizes across the nuclear membrane into the nucleus to regulate the expression of downstream genes [9]; researchers have found EA can inhibit the phosphorylation of JAK1, JAK2, and STAT1 to block this inflammatory response pathway [42].

## 4. Effects of EA in Neurological Diseases

Aging is associated with several diseases that threaten the health of older people, such as neurodegenerative diseases, which cause indelible and irreversible damage to both the physical and psychological well-being of older people and impose a heavy burden on their families and society. In recent years, a large number of studies have focused on the therapeutic effects of EA, demonstrating the health benefits of EA for neurodegenerative diseases. Three common neurological diseases are described here: AD, PD, and cerebral ischemia as well as the antioxidative and anti-inflammatory protective effects of EA.

### 4.1. EA in Alzheimer’s Disease

Alzheimer’s disease (AD) is one of the most common neurodegenerative diseases around the world, especially in the aging population. The patients behave with cognitive disorders, aphasia, and personality changes, which seriously affect patients’ quality of life [43]. The pathological change of AD can be concluded as three main features: extracellular amyloid-beta (Aβ) deposition, intracellular neurofibrillary tangles (NFT, caused by hyperphosphorylation of tau protein), and the loss of synapses and neurons.

Oxidative imbalance plays an important role in the pathogenesis of AD. The sources of free radicals in AD can be summarized in three aspects: (1) the accumulation of Fe and Al in neurons containing NFT can lead to the formation of OH·; (2) activated microglia that can form NO and $O_{2}^{-}$·; and (3), the abnormal metabolism of mitochondria can produce ROS [44]. It has been found that Fe, Zn, and Cu can bind to Aβ and APP and increase the oxidative stress of the AD brain [45]. Another pathologic feature of AD is the hyperphosphorylation of τ proteins, where the phosphorylation process is associated with the activation and oxidation of Nf-κB; this is important because the τ protein may be related to oxidative stress [46]. It has been clarified that EA can reduce the death of cells containing Aβ, lower the production of ROS, and mitigate DNA damage [47]. With its powerful free radical scavenging ability, EA can inhibit lipid peroxidation and reduce the production of NO, glycation end products, and so on [48].

In recent years, much evidence has proven that inflammation plays an important role in AD pathogenesis with the pathological observations of the inflammatory process [49]. As one of the pathogenetic features of AD, Aβ has been proven to activate the complement system and microglia [43]. Microglia are the immune cells in the CNS and the activation of microglia leads to the secretion of inflammatory cytokines, anaphylatoxin, free radicals, and neurotoxic substances which can kill neurons [43,50,51]. A recent study has shown that the activated microglia can synthesize proteolytic enzymes which can destroy cell structures and lead to the dysfunction of neurons [52]. The T-cells can also be activated in such a process and release inflammatory cytokines such as IL-1α, IL-1β, and TNF-α [53]. The excessive release of inflammatory cytokines induces the production of APP, which will positively upregulate the production of Aβ [51]. The continuous activation of pro-inflammatory responses could cause damage to the brain and the worsening of AD [52]. A study revealed that EA can regulate the production of IL-1β and TNF-1α to protect the damaged neurons induced by arsenic and improve the behavior and pathological change of AD [54]. Another key finding indicates that EA can reduce the number of microglia and inhibit the production of inflammatory cytokines [55] (Figure 4).

Plenty of studies focused on the Aβ and τ protein have shown the effects of EA. Researchers found that a group treated with EA exhibited a reduced level of Aβ plaques [56,57] and EA can significantly inhibit the hyperphosphorylation of the τ protein in animals’ hippocampus [57]. As a natural substance, EA is a potential therapeutic prospect for AD treatment.

### 4.2. EA in Parkinson’s Disease

PD is recognized as the most common neurodegenerative disorder after AD. The motor features of PD include bradykinesia, muscular rigidity, and resting tremors. The non-motor features include olfactory dysfunction, cognitive impairment, psychiatric symptoms, and autonomic dysfunction. The specific degeneration of dopaminergic neurons in the substantia nigra and the presence of Lewy bodies (LB) have been recognized microscopically [58]. It has been proven that oxidative stress is one of the primary underlying reasons for substantia nigra pars compacta loss, and neuroinflammation has been recognized as a culprit contributing to PD development [59].

More and more evidence indicates that oxidative stress is a key driver of dopaminergic neurodegeneration in all forms of PD [60,61] and ROS accumulation results in damage to the structural integrity and neuronal dysfunction [62]. Moreover, several epidemiological studies indicate that the use of anti-inflammatory medications reduces the risk of PD [63]. Other studies also revealed that inflammation can directly or indirectly contribute to the development and progression of PD [64]. There is a vicious circle between PD, redox imbalance, and chronic inflammation. As people age, excessive oxidative stress or brain injury can lead to microglial activation [65], and the activated microglia secrete pro-inflammatory factors such as TNF-α, IL-1β, and IL-18 [66]. The accumulation of pro-inflammatory factors leads to the continuing loss of dopamine neurons. The damaged dopamine neurons release neurotoxic factors which induce a secondary activation of microglia that further secrete pro-inflammatory factors leading to more loss of dopamine neurons, thus creating a vicious cycle leading to neuroinflammation and neurodegeneration [67].

Several studies have shown that EA plays an active role in PD by enhancing the antioxidant defense and reducing oxidative stress [68,69,70]. EA can reduce oxidative damage by regulating the Nrf2 and Nf-κB pathways and improving the activity of antioxidant enzymes and antioxidants [71]. It has also been indicated that EA can lead to a decreased malondialdehyde (MDA) level and increased activities of total glutathione (GSH), catalase, and SOD in the mouse model of PD [68]. Pretreatment with EA in the PD rats reduced ROS and improved the level of monoamine oxidase B (MAO-B), Nrf2, and HO-1. Additionally, after using the antagonist, the improved results disappeared [69]. In another study, after the oral administration of EA, the medial forebrain bundle (MFB)-injured rats appeared to recover their motor deficiencies by a significant increase in glutathione peroxidase (GPx) and SOD activity and decreasing the level of MDA [70]. The above evidence has shown that EA can protect dopamine neurons by inhibiting oxidative stress in the animal model of PD.

EA can also inhibit the release of endogenous inflammatory mediators in the animal model of PD, such as COX-2, iNOS, and cytokines [68], and has been confirmed to reduce MFB lesions and decrease inflammatory biomarkers such as TNF-α and IL-1β [72]. Furthermore, EA can reduce the activation of NLRP3 signaling associated with neuroinflammation [72] and the secretion of pro-inflammatory factors in microglia and has a neuroprotective effect against LPS-induced dopamine neuronal damage as well [73].

PD threatens our society with a substantial health and economic burden. To slow down or halt disease progression, developing novel therapeutic strategies acting on the underlying disease pathogenesis has become an urgent necessity. Based on the effects of EA on oxidative stress and chronic inflammation in the progress of PD, EA may be a potential drug in clinical therapy.

### 4.3. EA in Cerebral Ischemia

Cerebral ischemia/reperfusion is a kind of serious cerebrovascular disease with a high morbidity and fatality rate. This ischemia can induce hypoxia in brain tissue and lead to a series of oxidative stress and inflammatory responses, ultimate dysfunction, and even death of the neuron [74].

When cerebral ischemia happens, some enzymes such as succinate dehydrogenase and NADPH oxidase in the cellular mitochondria produce more ROS. The accumulation of ROS could directly injure the intracellular structure and induce neuronal apoptosis [75]. On the other hand, cerebral ischemia induces NOS to produce NO, and excess NO can react with the superoxide anion to produce peroxynitrite and interfere with the SOD activity [76].

The accumulation of inflammatory cells and cytokines is another feature of cerebral ischemia. Exogenous mononuclear phagocytes, T lymphocytes, and polymorphonuclear leukocytes can all trigger an inflammatory response in CNS [77]. If the inflammatory response lasts for a long period of time, it can induce neuroglial cell proliferation and brain atrophy [77]. It has been seen that IL-6 and TNF-α increase significantly in cerebral ischemia and inhibiting the IL-6 and TNF-α pathways can alleviate cerebral ischemia injury [78]. The increased expression of NLRP1 and NLRP3 is also related to cerebral ischemia injury [79,80]. Researchers have shown that NLRP1 and NLRP3 contribute to energy depletion, acidosis, tissue protease release, and increased ROS production [80]. However, the mechanisms of the activation of NLRP1 and NLRP3 receptors during cerebral ischemia remain unknown.

EA can scavenge free radicals generated during cerebral ischemia, increase the activity of antioxidant enzymes, and inhibit the expression of inflammatory factors such as IL-1β and TNF-α by activating the MAPK and Nf-κB pathways to reduce cell death [81]. In addition, the other roles of EA in cerebral ischemia include: (i) improving the decreased blood-brain barrier (BBB) permeability after cerebral ischemia by inhibiting the release of inflammatory cytokines [81,82] and (ii) increasing the activity of lactate dehydrogenase and reducing the damage of lactic acid accumulation to cells [82]. 

EA has protective effects on the ischemic brain, and as an antioxidant and anti-inflammatory natural substance, the side effects are low. However, the exact mechanism of its action still needs further investigation.

## 5. Conclusions

EA is a polyphenolic compound widely distributed in fruits and vegetables. From the analysis of its chemical structure, the four hydroxyl groups and two lactone functional groups are responsible for its active antioxidant and anti-inflammation effects. According to the results of numerous studies, EA is also a neuroprotective agent that can enhance neuronal function and memory and reduce damage to brain areas. Experimental evidence from cytology and histopathology has shown that EA has therapeutic potential for some neurological disorders such as AD, PD, and cerebral ischemia. EA can scavenge free radicals in the brain, increase the activity of antioxidative enzymes, and reduce the production of ROS, while its anti-inflammatory mechanism is manifested in the regulation of the Nf-κB, MAPKs, and JAK/STAT pathways to reduce the expressions of inflammatory factors such as TNF-α, IL-1β, and IL-6. 

The main problem of EA is its poor water solubility and low intestinal absorption. To overcome the low bioavailability of EA, researchers have developed several methods, such as EA-phospholipid complexes, polymer-based nanoparticles, nanomedicines (thermosensitive liposomes), and nano-sized metal cages [5]. We also suggest that random studies are needed to confirm the laboratory data to ensure the rigor and scientificity of the experiment. However, there are few clinical-based studies on EA and the protocols of clinical trials are diverse. More pharmacokinetic trials are needed, especially in humans, and further research focusing on the mechanisms of the neuronal protective effects of EA is needed as well. Furthermore, clinical studies evaluating the safety of EA and comparing its therapeutic efficacy with the approved drugs applied in clinical practice are essential. In summary, we are optimistic about the prospects for the clinical application of EA based on the current experimental data and results.

## Figures and Tables

**Figure 1 ijms-23-10937-f001:**
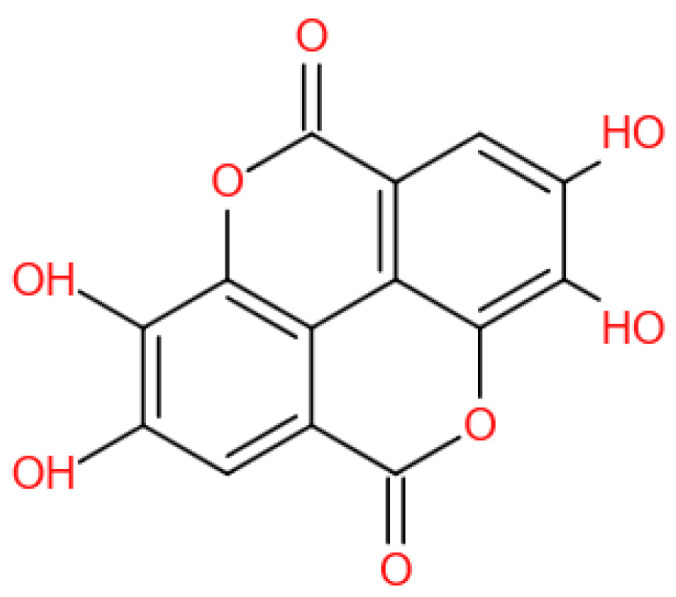
The chemical structure of EA.

**Figure 2 ijms-23-10937-f002:**
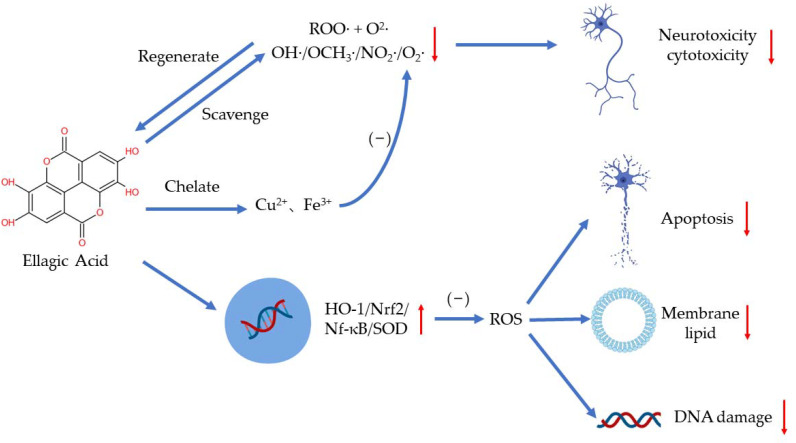
The antioxidant effects of EA.

**Figure 3 ijms-23-10937-f003:**
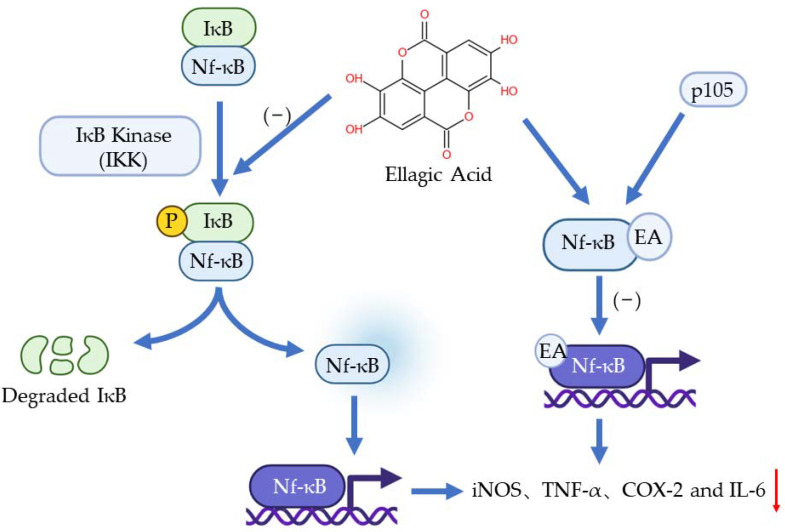
The anti-inflammation effects of EA to Nf-κB pathway.

**Figure 4 ijms-23-10937-f004:**
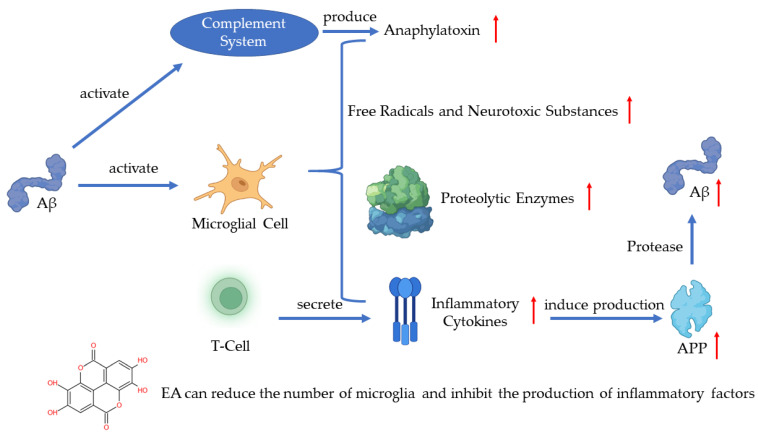
Pathogenesis of AD and the role of EA.

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
