# Peer review of "Ellagic Acid and Its Anti-Aging Effects on Central Nervous System"

_ijms, 2022, doi:10.3390/ijms231810937_

Round 1

Reviewer 2 Report

Congratulations to the authors for their work. The manuscript is well structured, and the images are clear and comprehensible. I only suggest emphasising that randomised studies are needed to confirm the laboratory data.

Reviewer 3 Report

This review article by Zhu and colleagues well describes the anti-aging effect of Ellagic acid. The article is well written, includes appropriates sections and references. 

Author Response

Dear reviewer:

Thanks very much for taking your time to review this manuscript. Your encouragement has given us great motivation to work. Hope you have a nice day! Thank you.